# CROSS-ATTENTION IS HALF EXPLANATION IN SPEECH-TO-TEXT MODELS

## ABSTRACT

Cross-attention is a core mechanism in encoder-decoder architectures, widespread in many fields, including speech-to-text (S2T) processing. Its scores have been repurposed for various downstream applications–such as timestamp estimation and audio-text alignment–under the assumption that they reflect the dependencies between input speech representation and the generated text. While the explanatory nature of attention mechanisms has been widely debated in the broader NLP literature, this assumption remains largely unexplored within the speech domain. To address this gap, we assess the explanatory power of cross-attention in S2T models by comparing its scores to input saliency maps derived from feature attribution. Our analysis spans monolingual and multilingual, single-task and multi-task models at multiple scales, and shows that attention scores moderately to strongly align with saliency-based explanations, particularly when aggregated across heads and layers. However, it also shows that cross-attention captures only about 50% of the input relevance and, in the best case, only partially reflects how the decoder attends to the encoder's representations–accounting for just 52-75% of the saliency. These findings uncover fundamental limitations in interpreting cross-attention as an explanatory proxy, suggesting that it offers an informative yet incomplete view of the factors driving predictions in S2T models.

## 1 INTRODUCTION

Cross-attention (Bahdanau et al., 2015) is the core mechanism of the encoder-decoder Transformer architecture (Vaswani et al., 2017), a model that has become foundational across numerous AI domains (Galassi et al., 2021; Lin et al., 2022; Lee et al., 2023; Wang et al., 2024b; Lu et al., 2024), including natural language processing (NLP). Designed for modeling dependencies between the generated output sequence and the input representations, the cross-attention scores–derived from the attention mechanism–have been leveraged in various NLP tasks (Hu, 2020; Zhang & Kim, 2023), such as source-target textual alignment (Garg et al., 2019; Chen et al., 2020), co-reference resolution (Voita et al., 2018), and word sense disambiguation (Tang et al., 2018).

In speech-to-text (S2T) modeling, cross-attention scores have been widely repurposed for diverse downstream applications such as audio-text alignment (Zhao et al., 2020; Lee et al., 2020), speaker identification (Kim et al., 2019), timestamp estimation (Li et al., 2022; Louradour, 2023; Zusag et al., 2024), and guiding simultaneous automatic speech recognition (ASR) and speech translation (ST) (Wang et al., 2024a; Papi et al., 2023a;b). These applications rely on the implicit assumption that cross-attention reliably indicates what the model attends to in the input signal during output generation. However, despite its widespread use, this assumption has never been verified. A key concern is that cross-attention operates over the encoder's output sequence–rather than directly on the raw audio–which may have been reorganized or mixed with contextual information. This phenomenon, known as context mixing (Mohebbi et al., 2023b), can potentially obscure the alignment between cross-attention weights and the original input signal. Similar concerns have been extensively debated in the NLP community, where the reliability of attention mechanisms as explanations has been both challenged and defended, leading to conflicting perspectives and empirical evidence (Serrano & Smith, 2019; Jain & Wallace, 2019; Wiegreffe & Pinter, 2019; Bastings & Filippova, 2020). In contrast, this question remains largely underexplored in the speech domain. Existing work on explainability in S2T has primarily focused on self-attention (Shim et al., 2022; Audhkhasi et al.,

2022; A Shams et al., 2024), or on empirically measuring the effects of context mixing (Mohebbi et al., 2023a), without directly assessing the explanatory potential of cross-attention mechanisms.

To address this gap, we present the first systematic analysis of cross-attention as a proxy for input-output dependencies in S2T models. Our study serves two main objectives: *i)* assessing the validity of using cross-attention as a surrogate for input-output alignment, and *ii)* evaluating whether it provides insights comparable to formal explainability methods such as feature attribution–while being more lightweight and less computationally expensive to obtain (Samek et al., 2021; Madsen et al., 2022). We compare cross-attention scores with input saliency maps derived from SPES (Fucci et al., 2025), the current state-of-the-art feature-attribution method in S2T, to determine the extent to which cross-attention captures which input features are relevant for models' predictions. In addition, we compute saliency maps on encoder outputs and compare them with cross-attention scores to evaluate whether cross-attention fully explains how the decoder uses encoded representations, avoiding potential discrepancies of context mixing. Our analysis spans ASR and ST tasks across monolingual, multilingual, and multitask settings using state-of-the-art speech processing architectures (Gulati et al., 2020) at multiple scales. With consistent trends across different settings, we find that cross-attention exhibits moderate to strong correlations with input saliency maps and aligns more closely with encoder output representations, suggesting an influence of context mixing. However, our results also indicate that the overall explanatory power of cross-attention is limited–accounting for only ∼50% of input relevance and, at best, 52-75% of encoder output saliency. Our findings uncover fundamental limitations in interpreting cross-attention as an explanatory proxy, suggesting that it provides an informative yet incomplete view of the factors driving predictions in S2T models.

## 2 RELATED WORKS

**Explainability in Speech-to-Text.** Explainable AI (XAI) has emerged to make model behavior more interpretable to humans, thereby supporting informed decision-making and responsible deployment (Barredo Arrieta et al., 2020). While XAI research has seen a rapid growth in the last years across multiple modalities, including vision and language (Sharma et al., 2024), progress in the speech domain has lagged. This gap arises from the inherent complexities of speech processing, including the multidimensional nature of speech signals across time and frequency, and the variability in output sequence length (Wu et al., 2024). Despite these challenges, growing concerns about trustworthiness are driving explainability efforts in speech classification (Becker et al., 2024; Pastor et al., 2024) and S2T generation (Mandel, 2016; Kavaki & Mandel, 2020; Trinh & Mandel, 2020; Markert et al., 2021; Wu et al., 2023; 2024; Fucci et al., 2025). Most of these works rely on perturbation-based methods that assess how input modifications affect model predictions (Covert et al., 2021b; Ivanovs et al., 2021). Among these, Fucci et al. (2025) recently proposed a technique for autoregressive S2T models that identifies regions of the spectrogram that most influence predictions to generate saliency maps. However, XAI methods are generally computationally expensive–especially perturbation-based approaches applied to large models (Luo & Specia, 2024; Yin et al., 2025)–which motivates exploring whether cross-attention, already computed at inference time, could serve as a lightweight alternative in a landscape still lacking efficient explainability tools for speech-based models.

**Attention as Explanation.** Attention mechanisms have been widely used to probe model behavior in text-based NLP, as attention scores often align with human intuitions about relevance and salience (Clark et al., 2019; Ferrando et al., 2024). Early studies proposed norm-based analyses to improve the interpretability of attention weights (Kobayashi et al., 2020; 2021; Mohebbi et al., 2021; Ferrando et al., 2022b), while others suggested aggregating attention across layers and heads to quantify input-output influence more systematically (Abnar & Zuidema, 2020; Ye et al., 2021; Chefer et al., 2021). While some have raised concerns about whether attention reliably reflects which inputs are actually responsible for outputs (Jain & Wallace, 2019; Serrano & Smith, 2019; Bastings & Filippova, 2020), others have proposed conditions under which attention can meaningfully explain model behavior (Wiegreffe & Pinter, 2019). More recent work highlights that attention aggregation may obscure localized, token-specific interactions (Modarressi et al., 2023; Yang et al., 2023; Oh & Schuler, 2023), motivating hybrid approaches that combine attention with other XAI techniques, such as attribution methods (Modarressi et al., 2022), or the use of attention as a regularization signal during interpretability-driven training (Xie et al., 2024). Despite the ongoing efforts, most research has focused on self-attention within encoders, with limited attention to feed-forward dynamics (Kobayashi et al., 2024) and even less to encoder-decoder models. A few studies have investigated

attention in encoder-decoder architectures (Nguyen et al., 2021), including in machine translation (Zhou et al., 2024), but cross-attention remains largely underexplored in the speech domain and has been absent from the broader "attention as explanation" debate in NLP. Our work seeks to bridge this gap by bringing cross-attention of S2T models into this broader conversation, aiming to assess whether it can serve as a reliable explanation–and where its limitations emerge.

## 3 METHODOLOGY

We assess the extent to which cross-attention scores (CA) explain how the model looks at input features when generating a token by comparing them to the saliency map on the input $\mathbf{SM}^X$, obtained with the state-of-the art feature-attribution method for S2T, SPES (Fucci et al., 2025). Additionally, to assess whether cross-attention more accurately reflects how the decoder accesses encoded representations–rather than capturing the model's full input-output behavior–we compare CA with the encoder-output saliency map $\mathbf{SM}^H$. By analyzing how the correlation between CA and $\mathbf{SM}^H$ deviates from that with $\mathbf{SM}^X$, we can indirectly quantify the impact of context mixing in the resulting explanations. In the following, we first discuss how we extract CA scores (Section 3.1), then how we compute $\mathbf{SM}^X$ and $\mathbf{SM}^H$ (Section 3.2), and how we compare them (Section 3.3).

### 3.1 CROSS-ATTENTION IN SPEECH-TO-TEXT

In S2T models, the cross-attention mechanism enables each decoder token to integrate relevant portions of the encoded speech features, thereby conditioning generation on the entire input.

Let $\mathbf{X} \in \mathbb{R}^{T \times F}$ denote the speech input represented by mel-spectrogram features, where $T$ is the number of time frames and $F$ the number of frequency bins. The encoder processes $\mathbf{X}$ into a sequence of hidden representations $\mathbf{H} = \text{Encoder}(\mathbf{X}) \in \mathbb{R}^{T' \times D}$, where $T' < T$ reflects the number of encoder time steps after subsampling with a factor of $s$[1] and $D$ is the hidden dimensionality. The decoder then autoregressively generates an output sequence $\mathbf{y} = (y_0, y_1, \ldots, y_I)$ of length $I$, where each token $y_i$ is predicted based on the previously generated tokens $y_{<i}$ and the encoder output $\mathbf{H}$.

At each decoder layer $\ell \in \{1, \ldots, L\}$, cross-attention scores are computed via dot-product attention (Graves et al., 2014) between the decoder's current hidden states $\mathbf{B}^{(\ell)} \in \mathbb{R}^{I \times D}$ and the encoder outputs $\mathbf{H}$. Specifically, the decoder states are linearly projected to queries $\mathbf{Q}^{(\ell)} = \mathbf{B}^{(\ell)} \mathbf{W}_Q^{(\ell)} \in \mathbb{R}^{I \times d_k}$, while the encoder outputs are projected to keys $\mathbf{K}^{(\ell)} = \mathbf{H} \mathbf{W}_K^{(\ell)} \in \mathbb{R}^{T' \times d_k}$ using learned projection matrices $\mathbf{W}_Q^{(\ell)}, \mathbf{W}_K^{(\ell)}$. The resulting cross-attention matrix CA is:

$$\mathbf{CA}^{(\ell)} = \text{softmax}\left(\frac{\mathbf{Q}^{(\ell)} \mathbf{K}^{(\ell)\top}}{\sqrt{d_k}}\right) \in \mathbb{R}^{I \times T'}$$

where each row $\mathbf{CA}_i^{(\ell)}$ represents the attention distribution over encoder time steps for the generation of output token $y_i$ at layer $\ell$. To capture diverse patterns, Transformer-based models employ multi-head attention (Vaswani et al., 2017). Each head $h \in \{1, \ldots, H\}$ uses separate learned projections:

$$\mathbf{Q}_h^{(\ell)} = \mathbf{B}^{(\ell)} \mathbf{W}_{Q,h}^{(\ell)}, \quad \mathbf{K}_h^{(\ell)} = \mathbf{H} \mathbf{W}_{K,h}^{(\ell)}$$

where $\mathbf{W}_{Q,h}^{(\ell)} \in \mathbb{R}^{D \times d_k}$, and $\mathbf{W}_{K,h}^{(\ell)} \in \mathbb{R}^{D \times d_k}$. These projections are used to compute head-specific attention scores, yielding one attention matrix per head and layer: $\{\mathbf{CA}_h^{(\ell)}\}_{\ell=1, h=1}^{L, H}$.

Extracting the full set of scores provides a fine-grained view of how each output token in the generated hypothesis attends to the encoder's representations across all layers and heads. To derive a single layer-wise or head-wise attention distribution, we compute the mean of the attention matrices over a subset $\mathcal{S} \subseteq \{1, \ldots, L\} \times \{1, \ldots, H\}$ of layers and heads:

$$\overline{\mathbf{CA}}^{(\mathcal{S})} = \frac{1}{|\mathcal{S}|} \sum_{(\ell, h) \in \mathcal{S}} \mathbf{CA}_h^{(\ell)} \in \mathbb{R}^{I \times T'}$$

---

[1]As the length of the speech inputs is, in general, $10\times$ longer that of the corresponding textual input, it is a common practice in S2T modeling to downsample the input through convolutional modules (Wang et al., 2020).

By selecting different index sets $\mathcal{S}$, this formulation yields layer-wise, head-wise, or global averages. For example, setting $\mathcal{S} = \{(\ell, h) : h = 1, \ldots, H\}$ gives the average across heads at a given layer $\ell$; $\mathcal{S} = \{(\ell, h) : \ell = 1, \ldots, L\}$ averages across layers for head $h$; and $\mathcal{S} = \{1, \ldots, L\} \times \{1, \ldots, H\}$ computes the full average. This averaged attention provides a more aggregated view of the model's attention patterns at a specific layer or attention head, summarizing how the model attends to the input speech over time.

### 3.2 FEATURE ATTRIBUTION FOR SPEECH-TO-TEXT

To better understand how S2T models associate individual output tokens with specific regions of the raw speech input or of the model's inner representations (e.g., the encoder output), we employ *feature-attribution* techniques that produce token-level *saliency maps*. These maps quantify the relevance of different portions of the input sequence in determining the model's predictions.

**Input Saliency Maps.** Let again $\mathbf{X} \in \mathbb{R}^{T \times F}$ denote a mel-spectrogram input, where $T$ is the number of time frames and $F$ the number of frequency bins, and $\mathbf{y} = (y_0, y_1, \ldots, y_I)$ the sequence of length $I$ of the autoregressively-generated tokens predicted based on the input and the previously generated tokens $y_{<i}$. To attribute the prediction of each token $y_i$ to specific parts of the input spectrogram, we adopt SPES (Fucci et al., 2025), the state-of-the-art feature-attribution method designed for autoregressive S2T modeling. SPES assigns a saliency score to each time-frequency element of $\mathbf{X}$, producing a saliency map $\mathbf{SM}_i^X \in \mathbb{R}^{T \times F}$ for each token $y_i$, where higher values indicate greater relevance of the corresponding time-frequency regions. SPES operates by clustering spectrogram elements based on energy profiles–capturing acoustic components such as harmonics and background noise–and estimating the influence of each cluster by perturbing it with probability $p_X$, repeated $N_X$ times. The effect of each perturbation–i.e., masking parts of the input with 0 values–is measured by computing the Kullback-Leibler (KL) divergence (Kullback & Leibler, 1951) between the model's original output distribution $P(y_i \mid y_{<i}, \mathbf{X})$ and the distribution resulting from the perturbed input $P^{(n)}(y_i \mid y_{<i}, \tilde{\mathbf{X}}^{(n)})$ at time $n \in \{1, \ldots, N_X\}$:

$$\mathrm{KL}_i^{(n)} = \mathrm{KL}\left(P(y_i \mid y_{<i}, \mathbf{X}) \,\|\, P^{(n)}(y_i \mid y_{<i}, \tilde{\mathbf{X}}^{(n)})\right)$$

The divergence scores are then mapped back to the corresponding cluster positions in the spectrogram and aggregated to form the token-specific saliency map $\mathbf{SM}_i^X \in \mathbb{R}^{T \times F}$. Stacking all saliency maps across the output sequence $\mathbf{y}$ yields a 3D saliency map:

$$\mathbf{SM}^X = (\mathbf{SM}_0^X, \mathbf{SM}_1^X, \ldots, \mathbf{SM}_I^X) \in \mathbb{R}^{I \times T \times F}$$

where each slice $\mathbf{SM}_i^X[t, f]$ quantifies the contribution of the spectrogram bin at time $t$ and frequency $f$ to the generation of token $y_i$.

**Encoder Output Saliency Maps.** We further examine the influence of the encoder's internal representations on the prediction of each output token. Let again $\mathbf{H} = \mathrm{Encoder}(\mathbf{X}) \in \mathbb{R}^{T' \times D}$ denote the sequence of encoder hidden states or *encoder output*, where $T'$ is the subsampled time dimension and $D$ is the hidden dimension. To assess the importance of the encoder output representations, we compute token-specific saliency maps $\mathbf{SM}_i^H \in \mathbb{R}^{T' \times 1}$, where each entry reflects the contribution of the corresponding hidden state to the generation of $y_i$. Each encoder state $\mathbf{H} = (\mathbf{h}_1, \ldots, \mathbf{h}_{T'})$ is perturbed–i.e., all its features are set to 0–independently with probability $p_H$, and the process is repeated $N_H$ times. The KL divergence is computed for each perturbation between the original and perturbed output distributions:

$$\mathrm{KL}_i^{(n)} = \mathrm{KL}\left(P(y_i \mid y_{<i}, \mathbf{H}) \,\|\, P^{(n)}(y_i \mid y_{<i}, \tilde{\mathbf{H}}^{(n)})\right)$$

The divergence scores are aggregated across perturbation trials to form the final saliency map, following the same strategy of SPES for the input-level saliency maps:

$$\mathbf{SM}^H = (\mathbf{SM}_0^H, \mathbf{SM}_1^H, \ldots, \mathbf{SM}_I^H) \in \mathbb{R}^{I \times T'}$$

where $\mathbf{SM}^H$ captures the temporal relevance of the encoder's internal sequence representations for each output token $y_i$.

### 3.3 CORRELATION

Since our focus lies in the temporal dynamics of the input $\mathbf{X}$, we aggregate the 3D saliency scores $\mathbf{SM}^X \in \mathbb{R}^{I \times T \times F}$ across the frequency dimension and downsample the time axis to produce a compressed representation $\mathbf{SM}^X \in \mathbb{R}^{I \times T'}$ compatible with the cross-attention granularity, where $T'$ corresponds to the number of encoder time steps. The aggregation is performed by taking the maximum saliency value over the frequency axis and within each corresponding time window. The resulting saliency map of each token $\mathbf{SM}_i^X \in \mathbb{R}^{T' \times 1}$ reflects the temporal relevance of the input spectrogram with respect to the generation of token $y_i$. Complementary experiments on the choice of the aggregation function are presented in Appendix A. Both $\mathbf{CA}$ and $\mathbf{SM}$ representations are normalized before computing the correlation scores, and the beginning and end of sentence are removed as they are not relevant for the analysis. The $\mathbf{CA}$ matrix is normalized frame-wise using mean-variance normalization to mitigate the impact of potential attention sinks at initial or final tokens (Clark et al., 2019; Ferrando et al., 2022a; Papi et al., 2023a; Xiao et al., 2024) on the correlation computation. Both $\mathbf{SM}^X$ and $\mathbf{SM}^H$ are normalized along the token dimension using the strategy proposed by Fucci et al. (2025), as saliency scores can vary widely across tokens due to differences in the original output distributions used to compute the KL divergence.

Following prior work on cross-attention matrices (Vig & Belinkov, 2019) and explainable AI (Eberle et al., 2023), we use Pearson correlation to quantify the relationship between cross-attention scores and saliency-based explanations. Pearson correlation is preferred over Kendall and Spearman because saliency scores are continuous, and their magnitude–not just ranking–is crucial. Rank-based measures are overly sensitive to small fluctuations among non-important features with near-zero scores, while Pearson better captures whether features are identified as important (high score) or not (low score). Specifically, given the two representations $\mathbf{CA}, \mathbf{SM} \in \mathbb{R}^{I \times T'}$, we compute the Pearson correlation coefficient $\rho$ to assess the similarity of their attribution patterns across output tokens and time steps. We first flatten each matrix into a vector of size $I \cdot T'$:

$$\mathbf{ca} = \text{vec}(\mathbf{CA}), \quad \mathbf{sm} = \text{vec}(\mathbf{SM}), \quad \mathbf{ca}, \mathbf{sm} \in \mathbb{R}^{I \cdot T'}$$

Then, the Pearson correlation coefficient $\rho \in [-1, 1]$ is computed as:

$$\rho(\mathbf{CA}, \mathbf{SM}) = \frac{\sum_{k=1}^{I \cdot T'} (\mathbf{ca}_k - \overline{\mathbf{ca}})(\mathbf{sm}_k - \overline{\mathbf{sm}})}{\sqrt{\sum_{k=1}^{I \cdot T'} (\mathbf{ca}_k - \overline{\mathbf{ca}})^2} \sqrt{\sum_{k=1}^{I \cdot T'} (\mathbf{sm}_k - \overline{\mathbf{sm}})^2}}$$

where $\overline{\mathbf{ca}}$ and $\overline{\mathbf{sm}}$ denote the means of vectors $\mathbf{ca}$ and $\mathbf{sm}$, respectively:

$$\overline{\mathbf{ca}} = \frac{1}{I \cdot T'} \sum_{k=1}^{I \cdot T'} \mathbf{ca}_k, \quad \overline{\mathbf{sm}} = \frac{1}{I \cdot T'} \sum_{k=1}^{I \cdot T'} \mathbf{sm}_k$$

This scalar value quantifies the linear relationship between the two saliency maps, with values closer to 1 indicating a strong positive correlation, and values near 0 indicating no correlation.

## 4 EXPERIMENTAL SETTINGS

### 4.1 DATA

To avoid potential data contamination issues (Sainz et al., 2023), we train from scratch a monolingual ASR model and two-sized multitask (ASR and ST) and multilingual (English and Italian) models. Details about training data and process are presented in Appendix B. Being the only non-synthetic dataset supporting both tasks and language directions, we select EuroParl-ST (Iranzo-Sánchez et al., 2020) as the test set for our analyses. The test set covers both *en* and *it* ASR, and *en-it* and *it-en* ST. The *it/it-en* section consists of 1,686 segments, for a total of approximately 6 hours of audio, while the *en/en-it* section contains 1,130 segments, for a total of approximately 3 hours of audio.

### 4.2 MODEL

The models analyzed in the paper are all composed of a Conformer encoder (Gulati et al., 2020) and a Transformer decoder, as Conformer is the current state-of-the-art architecture for S2T processing

(Guo et al., 2021; Srivastava et al., 2022; Li & Doddipatla, 2023). The monolingual ASR model (`base`) is composed of 12 encoder layers and 6 decoder layers. Each layer has 8 attention heads, 512 as embedding dimension, and FFNs dimension of 2,048. The vocabulary is built using a SentencePiece unigram model (Kudo & Richardson, 2018) with size 8,000 trained on *en* transcripts. The resulting number of parameters is 125M. The multitask and multilingual models are of two sizes, `small` and `large`, the first having 12 encoder layers and 6 decoder layers and the latter having 24 encoder layers and 12 decoder layers. In both sizes, each layer has 16 attention heads, an embedding dimension of 1,024, and an FFN dimension of 4,096. The vocabulary is built using a SentencePiece unigram model with size 16,000 trained on *en* and *it* transcripts. Two extra tokens–<lang:en> and <lang:it>–are added to indicate whether the target text is in *en* or *it*. The resulting number of parameters is 474M for the `small` model and 878M for the `large` model. In all models, the Conformer encoder is preceded by two 1D convolutional layers with stride 2 and kernel size 5, resulting in a fixed subsampling factor $s$ of 4. The kernel size of the Conformer convolutional module is 31 for both the point- and depth-wise convolutions. The input audio is represented by 80 Mel-filterbank features extracted every 10 ms with a window of 25 ms.

### 4.3 EVALUATION PROCESS

**Hypothesis and Cross-Attention Generation.** For the hypothesis generation, we use beam search with a beam size of 5 and a no-repeat n-gram size of 5. The attention scores are extracted from layers or heads during the output generation. The ASR and ST quality scores of the hypotheses are presented in Appendix C. The inference is performed using a single NVIDIA A40 GPU (40GB RAM) with a batch size of 40,000 tokens and takes ∼2.5 minutes for `base`, ∼3-5.5 minutes for `small`, and ∼3-6.5 for `large`, depending on the source language.

**Explanation Heatmaps Generation.** Following the best configuration obtained in SPES (Fucci et al., 2025), we adopt the Morphological Fragmental Perturbation Pyramid (Yang et al., 2021) for clustering, which relies on Simple Linear Iterative Clustering or SLIC (Achanta et al., 2012), a k-means-based algorithm that groups elements according to spectral patterns. We use the default parameters; the threshold length in seconds is 7.50s, the SLIC sigma is 0, the compactness is 0.1, and the number of patches per second for the MFPP technique is [400, 500, 600]. For the choice of $p_X$ and $N_X$, we refer to the parameters used in (Fucci et al., 2025), setting $p_X = 0.5$ and $N_X = 20,000$. The quality of the input explanations is presented in Appendix C. For the choice of $p_H$ and $N_H$, we use the same number of iterations of $N_X$, i.e., $N_H = 20,000$, while the optimal occlusion probability $p_H$ is determined over the dev set, resulting in $p_H = 0.7$, whose experiments are reported in Appendix D. The inference is performed using a single NVIDIA A40 GPU (40GB RAM) and takes ∼27 hours for `base`, ∼3-4 days for `small` and ∼6-8 days for `large`, depending on the source language.

**Correlation Computation.** The Pearson $r$ correlation score is computed using the `scipy` implementation[2] and averaging across samples in the test set.

## 5 RESULTS

### 5.1 DOES CROSS-ATTENTION REFLECT INPUT-OUTPUT DEPENDENCIES?

In this section, we compare CA with input saliency maps $SM^X$, which serve as an external reference for measuring input relevance. Specifically, in Section 5.1.1, we analyze the `base` model across all levels of granularity. Then, in Section 5.1.2, we extend the analysis to additional models (`small` and `large`), languages (*en* and *it*), and tasks (ASR and ST).

### 5.1.1 HEAD-WISE AND LAYER-WISE CORRELATIONS

Table 1 reports the correlation scores for the monolingual English ASR model (`base`), considering cross-attention at the head level, layer level, and in aggregated form.

At the individual head level, correlations with saliency maps are generally low. This suggests that attention heads, when taken in isolation, only partially capture the model's dependency on the input and often encode noisy or inconsistent relevance signals. However, not all heads are equal: some,

---

[2]https://docs.scipy.org/doc/scipy/reference/generated/scipy.stats.pearsonr.html

| Layer/Head | $h=1$ | $h=2$ | $h=3$ | $h=4$ | $h=5$ | $h=6$ | $h=7$ | $h=8$ | $h$-AVG |
|---|---|---|---|---|---|---|---|---|---|
| $\ell=1$ | 0.076 | -0.021 | 0.096 | 0.037 | 0.042 | 0.053 | -0.020 | 0.094 | 0.111 |
| $\ell=2$ | 0.013 | 0.122 | 0.039 | 0.171 | 0.123 | 0.089 | 0.119 | 0.041 | 0.178 |
| $\ell=3$ | 0.455 | 0.404 | 0.348 | 0.386 | 0.263 | 0.136 | 0.248 | 0.246 | 0.443 |
| $\ell=4$ | 0.452 | 0.344 | 0.227 | 0.405 | 0.314 | 0.512 | 0.414 | 0.495 | 0.546 |
| $\ell=5$ | 0.508 | 0.463 | 0.466 | 0.521 | 0.518 | 0.374 | 0.502 | 0.485 | 0.578 |
| $\ell=6$ | 0.377 | 0.394 | 0.508 | 0.386 | 0.517 | 0.371 | 0.424 | 0.322 | **0.588** |
| $\ell$-AVG | 0.546 | 0.456 | 0.455 | 0.574 | 0.529 | 0.541 | 0.525 | 0.500 | 0.577 |

Table 1: Pearson $\rho$ correlation between layer-wise ($\ell$) and head-wise ($h$) cross-attention and the explanations for the monolingual ASR model on English (`base`). The layer/head average ($\ell$-/$h$-AVG) correlation is computed between the averaged cross-attention across layers/head ($\overline{\mathbf{CA}}^{(\ell)}/\overline{\mathbf{CA}}_h$) and the input explanations $\mathbf{SM}^X$. **Bold** indicates the highest correlation, underline indicates the highest layer-wise and head-wise correlation. Low to high values are **GREEN** , to **YELLOW** , to **PINK** .

especially in the upper layers (layers 4-6), exhibit relatively stronger correlations. Notably, **averaging across heads consistently outperforms selecting individual heads**, suggesting that, despite head-level sparsity and weak individual correlations, the collective information captured across heads reflects input relevance more effectively. Moving from heads to layers, we find a clearer picture. Averaging attention scores across all heads within each layer boosts correlation substantially, with layer 6 standing out as the most aligned with the saliency maps. This is followed closely by layer 5 and the average across all layers, indicating that the **last layers exhibit the highest alignment with input relevance**. These results reinforce the idea that deeper layers encode higher-level semantic or task-relevant features, a trend previously observed in Transformer-based models (Clark et al., 2019). Interestingly, while averaging across heads improves alignment, averaging across both heads and layers does not yield the overall best result, even if values are close. This indicates that not all layers contribute equally and that indiscriminate aggregation can dilute the relevance signal.

Overall, the results show that appropriately selected and aggregated cross-attention scores exhibit only a *moderate* to *strong* correlation with input saliency maps, reaching values up to 0.588. This provides an initial indication of the limited explanatory power of cross-attention weights, which we further examine under multilingual and multitask conditions in Section 5.1.2.

### 5.1.2 MULTITASK AND MULTILINGUAL CORRELATIONS

To assess the impact of multilingual and multitask training on the correlation between cross-attention scores and saliency maps, we evaluate the `small` and `large` models. Layer-wise results are shown in Table 2, while head-wise results are omitted due to the noisy behavior observed in Section 5.1.1.

| Lang. | | Model | $\ell=1$ | $\ell=2$ | $\ell=3$ | $\ell=4$ | $\ell=5$ | $\ell=6$ | $\ell=7$ | $\ell=8$ | $\ell=9$ | $\ell=10$ | $\ell=11$ | $\ell=12$ | $\ell$-AVG |
|---|---|---|---|---|---|---|---|---|---|---|---|---|---|---|---|
| | | | | | | | | | | | | | | | |
| | | | | | | | | | | *en* | | | | | |
| Source language | *en* | `small` | 0.142 | 0.205 | 0.428 | **0.639** | **0.639** | 0.614 | | | - | | | | 0.633 |
| | | `large` | 0.151 | 0.162 | 0.214 | 0.320 | 0.289 | 0.434 | 0.581 | 0.597 | 0.611 | 0.597 | 0.551 | 0.561 | **0.621** |
| | *it* | `small` | 0.147 | 0.193 | 0.327 | 0.476 | 0.482 | 0.465 | | | - | | | | **0.485** |
| | | `large` | 0.151 | 0.164 | 0.223 | 0.300 | 0.285 | 0.383 | 0.451 | 0.467 | 0.461 | 0.461 | 0.430 | 0.431 | **0.492** |
| | | | | | | | | | *it* | | | | | | |
| | *en* | `small` | 0.173 | 0.203 | 0.344 | 0.547 | 0.550 | 0.539 | | | - | | | | 0.549 |
| | | `large` | 0.168 | 0.176 | 0.235 | 0.300 | 0.306 | 0.413 | 0.514 | 0.526 | 0.529 | 0.529 | 0.513 | 0.516 | **0.551** |
| | *it* | `small` | 0.145 | 0.209 | 0.374 | 0.527 | 0.532 | 0.525 | | | - | | | | **0.539** |
| | | `large` | 0.169 | 0.157 | 0.215 | 0.324 | 0.297 | 0.407 | 0.501 | 0.503 | 0.516 | 0.518 | 0.479 | 0.482 | **0.544** |

Table 2: Person $\rho$ correlation between layer-wise cross-attention $\overline{\mathbf{CA}}^{(\ell)}$ and the input explanations $\mathbf{SM}^X$ for the multitask (ASR and ST) and multilingual (English and Italian) models (`small` and `large`). **Bold** indicates the highest overall correlation, underline indicates the highest correlation across layers. Low to high values are **YELLOW** to **AQUA** for ASR, and to **RED** for ST.

Across all configurations, we observe that *en* ASR yields the highest correlation values, outperforming even the monolingual `base` model (Section 5.1.1). This suggests that large-scale multilingual training

| | | Target Language | | | | | | | | | | | | |
| --- | --- | --- | --- | --- | --- | --- | --- | --- | --- | --- | --- | --- | --- | --- |
| **Lang.** | | *en* | | | | | | | | | | | | |
| | **Model** | $\ell=1$ | $\ell=2$ | $\ell=3$ | $\ell=4$ | $\ell=5$ | $\ell=6$ | $\ell=7$ | $\ell=8$ | $\ell=9$ | $\ell=10$ | $\ell=11$ | $\ell=12$ | $\ell$-AVG |
| *en* | base | 0.105 | 0.179 | 0.602 | 0.708 | 0.745 | 0.693 | | | | - | | | **0.752** |
| | small | 0.137 | 0.220 | 0.492 | **0.712** | 0.704 | 0.664 | | | | - | | | 0.706 |
| | large | 0.161 | 0.143 | 0.224 | 0.352 | 0.316 | 0.476 | 0.651 | 0.695 | 0.697 | 0.699 | 0.629 | 0.641 | **0.718** |
| *it* | small | 0.140 | 0.196 | 0.362 | 0.502 | 0.495 | 0.467 | | | | | | | **0.519** |
| | large | 0.131 | 0.128 | 0.207 | 0.304 | 0.287 | 0.407 | 0.511 | 0.553 | 0.541 | 0.545 | 0.491 | 0.509 | **0.633** |
| | | *it* | | | | | | | | | | | | |
| *en* | small | 0.176 | 0.235 | 0.439 | 0.655 | 0.650 | 0.606 | | | | - | | | **0.659** |
| | large | 0.184 | 0.160 | 0.241 | 0.348 | 0.357 | 0.484 | 0.637 | 0.667 | 0.661 | 0.668 | 0.625 | 0.625 | **0.683** |
| *it* | small | 0.133 | 0.199 | 0.397 | 0.579 | 0.588 | 0.555 | | | | - | | | **0.594** |
| | large | 0.152 | 0.119 | 0.187 | 0.322 | 0.303 | 0.423 | 0.560 | 0.605 | 0.606 | **0.619** | 0.573 | 0.565 | 0.573 |

Source language

Table 3: Person $\rho$ correlation between layer-wise cross-attention $\overline{\mathbf{CA}}^{(\ell)}$ and the encoder output explanations $\mathbf{SM}^H$ for all models (base, small and large). **Bold** indicates the highest overall correlation, underline indicates the highest correlation across layers. Low to high values for ASR are YELLOW to ORANGE, and ST are LIGHT CYAN to DARK CYAN.

enhances the alignment between cross-attention and saliency maps, likely due to the improved generalization capacity of the model. In contrast, *en-it* ST shows a drop in correlation, which is expected given the increased complexity of ST compared to ASR. When considering *it* as the source language, we observe a similar pattern: ASR correlations are consistently higher than ST, yet remain below their *en* counterparts. This discrepancy aligns with the data distribution in training, where *en* accounts for 84% of the data versus 16% for *it*, resulting in more robust representations for *en*. At the layer level, we find consistent evidence that the last decoder layers yield stronger correlations, reaffirming the trends observed in Section 5.1.1. The specific optimal layer varies with model size: layer 5 performs best in small, while layers 8-10 achieve the highest correlations in large. Nevertheless, correlation values across the last layers remain very close, suggesting that their cross-attention scores provide the most robust alignment with saliency maps across both tasks and languages. This trend is further supported by downstream application results, where the final layers have shown the best token-level performance (Papi et al., 2023a;b; Wang et al., 2024a).

Averaging attention scores across layers further improves the correlation with saliency maps in almost all configurations. The only exceptions are *en* and *it* ASR in small, where selective-layer extraction offers a marginal improvement (0.006 for English, 0.001 for Italian). Therefore, similarly to what we observed in Section 5.1.1, averaging attention across heads and layers consistently yields the best or near-best correlation with *moderate* to *strong* correlation with input saliency maps, even considering large-scale models trained in multitask and multilingual settings. Nonetheless, this alignment accounts for only 49-63% of the total input relevance, indicating that **cross-attention falls short of fully accounting for the S2T models' behavior**. Since this limitation may stem from the phenomenon of context mixing, in Section 5.2 we analyze the correlation between cross-attention and encoder output–representations that have already undergone transformation by the encoder–to better isolate the true explanatory power of cross-attention.

## 5.2 WHAT IS THE IMPACT OF CONTEXT MIXING?

While Section 5.1 focused on input relevance, we now investigate whether CA aligns more closely with encoder output saliency maps. A higher correlation with encoder output representations would support the hypothesis that discrepancies between cross-attention and input saliency arise from context mixing, due to the reorganization of information within the encoder. To this end, we compare CA with encoder output saliency maps $\mathbf{SM}^H$, which attribute relevance to the encoder hidden states for each output token (Section 3.2). Layer-wise results for all models are presented in Table 3.

Even when examining encoder output representations, we observe trends consistent with those identified in Section 5.1. Specifically, when averaged across decoder layers, cross-attention scores consistently provide the strongest or nearly optimal correlation with saliency maps, with the last decoder layers offering more representative explanations than the first ones across all models. As expected, correlation with encoder output representations consistently yields higher scores than

those obtained from input representations, with absolute $\rho$ differences ranging from 0.03 to 0.18, quantifying the influence of context mixing effects to 6.6-16.7%. The increased correlation is also visually evident in the example shown in Figure 1, where **CA** aligns more closely with the relevance scores from $\mathbf{SM}^H$ than with those from $\mathbf{SM}^X$. However, despite being unaffected by context mixing, the correlation between **CA** and $\mathbf{SM}^H$ remains limited–capturing only 52-75% of the relevance. **This gap underscores the inherent limitations in relying solely on cross-attention as an explanation mechanism, reinforcing its role as an informative but incomplete proxy for explainability in S2T models**–not only for input-level saliency, but even at the encoder-output level, where cross-attention directly operates.

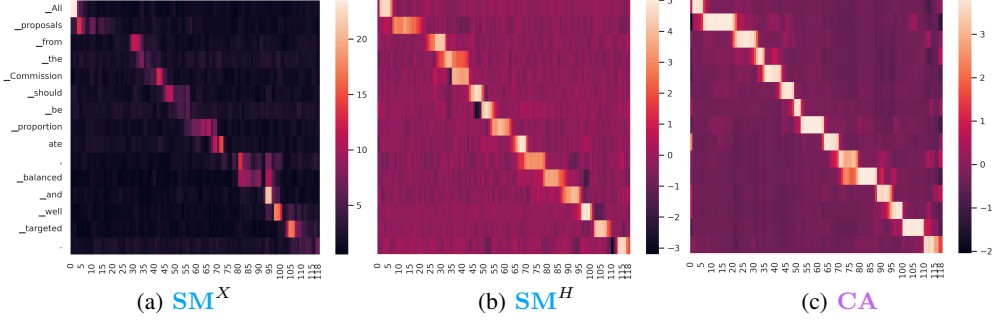

(a) $\mathbf{SM}^X$  (b) $\mathbf{SM}^H$  (c) **CA**

Figure 1: Input (a) and encoder output (b) saliency maps and cross-attention matrix (c) extracted from the dev set. The output is produced by the `base` model, more examples are available in Appendix E.

## 6 DISCUSSION AND CONCLUSIONS

**Discussion.** Our results demonstrate that, although **CA** scores moderately correlate with aggregated $\mathbf{SM}^X$ (with a correlation peaking around 0.45-0.55 in the best-performing settings), they consistently fall short of capturing the full input relevance–even when context mixing effects are factored out. To directly assess explanation quality, we compute the deletion metric (see Appendix A) on the `base` model, finding that **CA** achieves 41.2, compared to 52.9 for frequency-aggregated $\mathbf{SM}^X$ and 91.3 for full-resolution maps. This gap underscores that **CA** discards fine-grained time-frequency cues and produces weaker attributions, even under identical aggregation. As further discussed in Appendix F, our analysis is bounded by the use of SPES as the attribution baseline, but the consistent underperformance across correlation and deletion confirms that **CA** offers, at best, an incomplete picture of model behavior. These results also carry implications for downstream tasks. In applications such as timestamp prediction, prior work often relies on attention from a single decoder layer or head (Wang et al., 2024a; Papi et al., 2023a;b; Zusag et al., 2024). Our analysis suggests that averaging across layers and, especially, across heads provides a closer match to saliency behavior and could improve these methods. Building on past success with attention regularization in ASR (e.g., imposing monotonicity as in Zhao et al. 2020), similar training-time strategies–such as auxiliary losses that align attention with saliency–could further benefit downstream applications, enhancing both interpretability and task performance. In summary, **CA should not be treated as a stand-alone XAI tool**. It provides lightweight cues that may complement attribution-based methods, but it cannot replace them. Reframing **CA** as an auxiliary rather than a proxy recalibrates expectations and grounds future work on more faithful and effective approaches to explainability in S2T models.

**Conclusions.** We presented the first systematic analysis of cross-attention in S2T through the lens of explainable AI, comparing it to saliency maps across tasks, languages, and model scales. Cross-attention moderately to strongly aligns with saliency–especially when averaged across heads and layers–but captures only about half of the input relevance. Even when disentangling the effect of context mixing by analyzing encoder outputs, it explains just 52-75% of saliency. This gap reveals intrinsic limits of cross-attention as an explanation mechanism: it offers informative cues but only a partial view of the factors driving S2T predictions.

ETHIC STATEMENT

**Broader Implications.** Explainability in S2T systems has tangible implications for AI transparency, especially in high-stakes settings such as healthcare, legal transcription, and educational accessibility. Our findings provide insights about the usage of cross-attention as a tool for identifying how models relate output predictions to input regions, which can support auditing, debugging, and fair deployment. However, there is a risk that misinterpreted attention visualizations may be overtrusted by non-expert users, reinforcing false confidence in system behavior (Rudin, 2019). Moreover, our language choices and focus on high-resource speech still reflect global imbalances in language technology access (Joshi et al., 2020). Future work should extend this analysis to low-resource and underrepresented languages to promote broader inclusion.

**Use of Large Language Models.** For the writing process, ChatGPT was employed exclusively to correct grammar in content authored by humans.

REPRODUCIBILITY STATEMENT

To ensure the reproducibility of our results, we described in Section 4 all the details regarding our model training, training and evaluation data, and evaluation procedure. Moreover, we relied only on openly available data and on open source code[3] for the generation of the saliency maps. Lastly, all models (described in Section 4), code, attention scores, and explanation artifacts will be released under the Apache 2.0 (code) and CC-BY 4.0 (all other materials) licenses upon paper acceptance.

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

# A EFFECT OF AGGREGATION FUNCTIONS ON INPUT EXPLANATIONS

To properly obtain input-level explanations comparable with the dimensions of cross-attention scores (i.e., making $\mathbf{SM}^X \in \mathbb{R}^{I \times T'}$), we explore the effect of different aggregation strategies over the time and frequency dimensions.

To compare and select the best aggregation strategy, we adopt the *deletion* metric (Nauta et al., 2023), which quantifies the decline in prediction quality as the most relevant input frames–identified by the explanation–are progressively removed. Specifically, we adapt the implementation by Fucci et al. (2025) for S2T tasks, replacing the top-ranked time frames in the input spectrogram $\mathbf{X}$ with zero vectors in 5% increments, based on the aggregated saliency map $\mathbf{SM}^X$. Since $\mathbf{SM}^X$ operates on an aggregated time dimension $T'$, which is smaller than the original time dimension $T$ of $\mathbf{X}$, we upsample $T'$ to match $T$ using nearest-neighbor interpolation. Prediction quality is measured using the word error rate (WER), specifically the wer_max scorer from the SPES repository.[4] Lastly, we compute the area under the WER curve to quantify the faithfulness of each explanation method.

Table 4 reports the deletion scores and, for completeness, the Pearson $\rho$ correlations between the cross attention scores $\mathbf{CA}$ and the saliency maps $\mathbf{SM}^X$ for the representations aggregated following three strategies:

- **2D average pooling**, applied over the entire time-frequency plane to obtain its *average* value and computed through adaptive_avg_pool2d[5];
- **2-step pooling (1D maximum + 1D average)**, where max pooling is applied along the frequency axis, followed by averaging over time, and computed by applying max_pool1d[6] and avg_pool1d,[7] respectively;
- **2D maximum pooling**, applied over the entire time-frequency plane to obtain its *maximum* value and computed through adaptive_max_pool2d[8].

The aggregation functions were selected to contrast methods that either isolate the most relevant features (with maximum pooling) or represent their mean relevance (with average pooling). Similarly, the 2-step approach has been tried to first isolate relevance patterns in the frequency domain, a dimension that is not present in cross-attention representation, and then average across the time dimension to match the downsampled time resolution of the cross-attention scores.

| Aggregation | | $\ell = 1$ | $\ell = 2$ | $\ell = 3$ | $\ell = 4$ | $\ell = 5$ | $\ell = 6$ | $\ell$-AVG | ASR Del.↑ |
|---|---|---|---|---|---|---|---|---|---|
| *frequency* | *time* | | | | | | | | |
| 2D avg | | 0.090 | 0.142 | 0.355 | 0.434 | 0.457 | 0.466 | 0.459 | 53.03 |
| 1D max | 1D avg | 0.115 | 0.176 | 0.441 | 0.534 | 0.560 | 0.565 | 0.565 | 55.18 |
| 2D max | | 0.115 | 0.180 | 0.443 | 0.540 | 0.572 | 0.582 | **0.572** | **57.04** |

Table 4: Pearson $\rho$ correlation between layer-wise ($\ell$) cross-attention and the explanations, and the deletion scores (**ASR Del.**) for the different aggregation functions of the monolingual ASR model on English (base) on the dev set. The layer average ($\ell$-**AVG**) correlation is computed between the averaged cross-attention across layers ($\overline{\mathbf{CA}}^{(\ell)}$) and the input explanations $\mathbf{SM}^X$. **Bold** indicates the highest result, underline indicates the highest layer-wise correlation.

Among the tested methods, we observe that the 2D maximum pooling aggregation (2D max) yields the best quality explanations, obtaining the highest deletion score, while the 2D average pooling (2D avg) is the worst, with the lowest deletion score. Looking at the correlations, we notice that they follow the same trend of deletion scores, with the 2D max yielding the best $\rho$. In particular, 2D avg consistently has the lowest correlations compared to the 2D max, particularly in the last layers (e.g., 0.457 against 0.572 at layer 5). Regarding the 2-step pooling approach, we not only observe an improved deletion score but also better correlation scores compared to 2D avg, especially from layer

---

[4]https://github.com/hlt-mt/FBK-fairseq

[5]https://docs.pytorch.org/docs/stable/generated/torch.nn.functional.adaptive_avg_pool2d.html

[6]https://docs.pytorch.org/docs/stable/generated/torch.nn.functional.max_pool1d.html

[7]https://docs.pytorch.org/docs/stable/generated/torch.nn.functional.avg_pool1d.html

[8]https://docs.pytorch.org/docs/stable/generated/torch.nn.functional.adaptive_max_pool2d.html

3 onward, approaching the best performance with a layer-average correlation of 0.565. Nevertheless, the explanation quality is still lower compared to 2D max (i.e., 55.18 against 57.04), which also achieves the highest correlations at nearly every layer, peaking at 0.582 in layer 6, and yielding the best overall correlation among the averaged cross-attention across layers (i.e., 0.572).

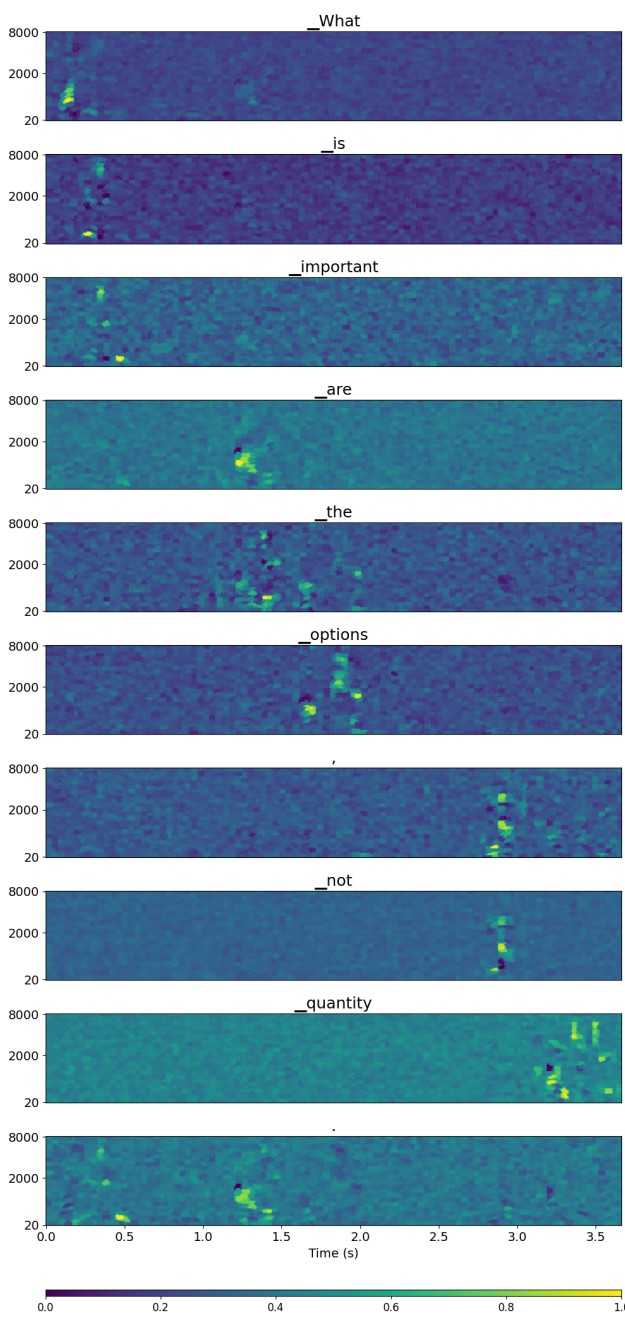

Figure 2: An example of $\mathbf{SM}^X$ maps for the predicted sentence "What is important are the options, not quantity". The frequency axis is represented in Hertz on a logarithmic scale.

These results indicate that global averaging over time and frequency may obscure localized salient regions, and this is particularly impactful in the frequency dimension, where preserving saliency seems to play a crucial role. This is due to the fact that key elements in the saliency maps are often well localized along the frequency axis. As shown in Figure 2, for all tokens saliency consistently concentrates in specific frequency bands. These bands are typically below 2000 Hz, where many

resonant frequencies for speech are found (Stevens, 2000). As a result, smoothing operations such as 2D average pooling–or, to a lesser extent, the 2-step approach–tend to blur these concentrated regions, thereby diluting the saliency. This observation motivates our choice to adopt 2D max pooling in the main experiments.

# B  TRAINING SETTINGS

## B.1  TRAINING DATA

For the monolingual ASR model, we leverage the speech-to-text English data available for the IWSLT 2024 evaluation campaign (offline task),[9] namely: CommonVoice (Ardila et al., 2020), CoVoST v2 (Wang et al., 2021b), Europarl-ST (Iranzo-Sánchez et al., 2020), LibriSpeech (Panayotov et al., 2015), MuST-C v1 (Di Gangi et al., 2019), TEDLIUM v3 (Hernandez et al., 2018), and VoxPopuli ASR (Wang et al., 2021a). The resulting training set is about 3k hours of speech.

For the multitask (ASR and ST) multilingual large-scale models, we leverage more than 150k hours of open-source speech[10] in English (*en*) and Italian (*it*), namely: CommonVoice, CoVoST v2, FLEURS (Conneau et al., 2023), MOSEL (Gaido et al., 2024a), MLS (Pratap et al., 2020), and YouTube-Commons[11] (from which 14.2k hours of *en* and 1.8k for *it* have been extracted). For datasets missing the translations, we generated them using `MADLAD-400 3B-MT` (Kudugunta et al., 2023). This setting allows us to verify our analysis with a large-scale setting similar to the scale of a popular model such as OWSM (Peng et al., 2023) and 2 times that of NVIDIA Canary (Puvvada et al., 2024) while having complete control of data used during training, ensuring that data contamination issues are avoided completely.

## B.2  TRAINING PROCESS

We train all models using a combination of three losses: *i)* a label-smoothed cross-entropy loss ($\mathcal{L}_{CE}$) applied to the decoder output using the target text as the reference (transcripts for ASR and translations for ST), *ii)* a CTC loss (Graves et al., 2006) computed using transcripts as reference ($\mathcal{L}_{CTCsrc}$) on the output of the 8$^{th}$ encoder layer for `base` and `small` and the 16$^{th}$ for `medium`, *iii)* a CTC loss on the final encoder output ($\mathcal{L}_{CTCtgt}$) applied to predict the target text (Yan et al., 2023). The final loss is the weighted sum of the above-mentioned losses:

$$\mathcal{L} = \lambda_1 \mathcal{L}_{CE} + \lambda_2 \mathcal{L}_{CTCsrc} + \lambda_3 \mathcal{L}_{CTCtgt}$$

where $\lambda_1, \lambda_2, \lambda_3 = 5.0, 1.0, 2.0$, and the label smoothing factor of the CE is $0.1$. The optimizer is AdamW with momentum $\beta_1, \beta_2 = 0.9, 0.98$, a weight decay of $0.001$, a dropout of $0.1$, and clip normalization of $10.0$.

The monolingual ASR `base` model is trained on all 3k hours of ASR data for 200k steps using Noam as the learning rate scheduler (Vaswani et al., 2017) with a peak of 2e-3 and 25,000 warm-up steps.

The multitask and multilingual models are trained using a two-stage approach, where the model is pre-trained first on ASR data only (ASR pre-training) and then trained on both ASR and ST data (ASR+ST training). For the ASR pre-training, the learning rate scheduler adopted for the `small` model is the same as the `base` model. For the `medium` model, we adopted a piece-wise warm-up on the Noam scheduler to avoid divergence issues (Peng et al., 2024), with the learning rate first increasing linearly to 2e-5 for 25k steps and then to 2e-4 for an additional 25k steps, followed by the standard inverse square root function. For the ASR+ST training, we sample the ASR target with probability 0.5 and use the ST target otherwise following the same settings of ASR pre-training, except for the learning rate that is set to a constant value of 1e-4 for `small` and 1e-5 for `medium`, following the same downscale of the ASR pre-taining. Both training stages lasted 1M steps, corresponding to ∼6 epochs over the training data.

All trainings are performed on fairseq-S2T (Wang et al., 2020). Following the default settings, we apply utterance-level Cepstral Mean and Variance Normalization (CMVN), SpecAugment (Park et al.,

---

[9]https://iwslt.org/2024/offline

[10]Speech, transcripts, and translations released under an open-source license such as CC-0 and CC-BY 4.0.

[11]https://hf.co/datasets/PleIAs/YouTube-Commons

2019), and filter out segments longer than 30 seconds to optimize memory requirements during all stages of the training.

For the `base` model, the trainings are executed on 4 NVIDIA A100 GPUs (64GB RAM) with a mini-batch of 40,000 tokens, an update frequency of 2, and averaging the last 7 checkpoints obtained from the training. For the multitask and multitlingual models, we use mini-batches of 10,000 tokens for the `small` and 4,500 for the `medium` with an update frequency of, respectively, 2 and 6 on 16 NVIDIA A100 GPUs (64GB RAM), save checkpoints every 1,000 steps and average the last 25 checkpoints to obtain the final models.

## C  QUALITY METRICS FOR THE REPORTED MODELS

The quality of the ASR hypotheses is evaluated with the WER metric using the jiWER library[12] and applying the Whisper text normalizer[13]. The quality of the ST hypotheses is evaluated using COMET (Rei et al., 2020) version 2.2.4, with the default model.[14] The quality of the explanations is obtained by measuring both *deletion* and *size* metrics available in the SPES repository, using `wer_max` as the scorer for ASR and `bleu` for ST, as described in (Fucci et al., 2025).

| Model | WER ↓ | | COMET ↑ | | ASR Del. ↑ | | ST Del. ↓ | | Size ↓ | | | |
|---|---|---|---|---|---|---|---|---|---|---|---|---|
| | en | it | en-it | it-en | en | it | en-it | it-en | en | it | en-it | it-en |
| Whisper | 10.6 | **9.0** | - | 0.797 | | - | | - | | | - | |
| Seamless | 11.3 | **9.0** | 0.795 | **0.813** | | - | | - | | | - | |
| OWSM v3.1 | 11.9 | 17.0 | 0.634 | 0.559 | | - | | - | | | - | |
| base | **9.5** | - | - | | 91.3 | - | | - | 29.7 | - | | |
| small | 11.7 | 22.3 | 0.854 | 0.754 | **92.6** | **97.0** | **2.4** | 2.4 | **29.4** | 28.2 | 30.0 | 29.4 |
| large | 11.1 | 21.7 | **0.862** | 0.765 | 90.8 | **97.0** | 2.7 | **2.3** | 30.6 | 30.5 | **29.8** | **28.7** |

Table 5: ASR and ST output quality (WER and COMET) and explanation quality (deletion and size) for all models analyzed in the paper on the EuroParl-ST test sets.

For comparison, in Table 5, we also report results obtained from popular large-scale models, namely Whisper (Radford et al., 2023), OWSM v3.1 (Peng et al., 2024), and SeamlessM4T (Barrault et al., 2023). Looking at the transcription/translation quality performance, we observe that both the monolingual `base` model and the multitask multilingual `small` and `large` models are mostly able to achieve competitive results, even outperforming the well-known models in two cases (*en* ASR for `base` and *en-it* ST for `large`). While our models and OWSM v3.1 strive to be on par on *it* with models with closed training data (Whisper and Seamless), they are able to close the gap on *en*, most probably given a larger availability of public training data. Moreover, the highest performance of `base` on *en* ASR compared to the `small` and `large` can be attributed to both the specialization of the model and the presence of the EuroParl-ST training set in the training data.

Moving to the explanation quality, we observe that both deletion and size scores are comparable across all three models analyzed in the paper and coherent with values obtained in the original SPES paper (Fucci et al., 2025) on different benchmarks and models. Overall, the deletion scores for ASR are close to the highest possible value (i.e., 100), especially on *it*, where 97% is achieved. Similarly, the deletion scores for ST are close to 0, indicating that the quality of explanations is very high. The size scores are all close, ranging between 28.2 and 30.6 among models, languages, and tasks, indicating a good compactness of the explanations.

## D  EFFECT OF OCCLUSION PROBABILITY ON ENCODER OUTPUT EXPLANATIONS

To properly choose the occlusion probability ($p_H$) for the encoder output explanations $\mathbf{SM}^H$, we conducted experiments by varying this probability in the set of $\{0.1, 0.3, 0.5, 0.7, 0.9\}$, similarly to what has been done for determining the input occlusion probability ($p_X$) in SPES (Fucci et al., 2025).

---

[12]https://pypi.org/project/jiwer/

[13]https://pypi.org/project/whisper-normalizer/

[14]https://hf.co/Unbabel/wmt22-comet-da

In Table D, we report the *deletion* metric computed on the dev set and, for completeness, the results of the Pearson $\rho$ correlation with cross-attention $\mathbf{CA}$. Analogously to the deletion metric computed on the input saliency maps (Fucci et al., 2025), we compute the deletion on the encoder output saliency maps by iteratively replacing portions of the encoder output sequence $\mathbf{H} = \{\mathbf{h}_1, \ldots, \mathbf{h}_I\}$ with zero vectors, and removing 5% of the most important time frames at each step based on their saliency. Frame importance is determined using saliency maps $\mathbf{SM}^H$ aggregated at the sentence level. The output quality is evaluated using the same `wer_max` scorer from the SPES repository. Lastly, we compute the area under the curve of the WER progression to quantify the faithfulness of the explanation.

| Occlusion Prob. | $\ell = 1$ | $\ell = 2$ | $\ell = 3$ | $\ell = 4$ | $\ell = 5$ | $\ell = 6$ | $\ell$-AVG | ASR Del. ↑ |
|---|---|---|---|---|---|---|---|---|
| $p_H = 0.9$ | 0.170 | 0.224 | 0.597 | 0.696 | 0.717 | 0.656 | 0.741 | 58.20 |
| $p_H = 0.7$ | 0.116 | 0.184 | 0.589 | 0.701 | 0.742 | 0.684 | **0.746** | **61.45** |
| $p_H = 0.5$ | 0.080 | 0.146 | 0.549 | 0.657 | 0.706 | 0.654 | 0.700 | 57.30 |
| $p_H = 0.3$ | 0.053 | 0.111 | 0.480 | 0.576 | 0.624 | 0.583 | 0.614 | 50.45 |
| $p_H = 0.1$ | 0.039 | 0.093 | 0.418 | 0.508 | 0.544 | 0.531 | 0.543 | 44.34 |

Table 6: Pearson $\rho$ correlation between layer-wise ($\ell$) cross-attention and the explanations, and deletion score (**ASR Del.**) by varying the occlusion probability $p_H$ of the monolingual ASR model on English (`base`) on the dev set. The layer average ($\ell$-AVG) correlation is computed between the averaged cross-attention across layers ($\overline{\mathbf{CA}}^{(\ell)}$) and the encoder output explanations $\mathbf{SM}^H$. For each $p_H$, we also report the deletion metric. **Bold** indicates the highest correlation, underline indicates the highest layer-wise correlation.

From the results, we can notice that higher occlusion probabilities yield not only a better deletion score but also an increased correlation with $\mathbf{CA}$. The overall best correlation is achieved when averaging across all layers, and layer 5 achieves the best layer-specific correlation values, a phenomenon that remains coherent even when varying the occlusion probability. Interestingly, the deletion scores and the $\mathbf{CA}$-$\mathbf{SM}^H$ correlations always follow the same trend, with the best values achieved with $p_H = 0.7$, which we used in all experiments reported in the main paper.

# E EXAMPLES

Examples of different saliency maps and cross-attention representations obtained with the `large` model are presented in Figure 3.

We notice similar relevance patterns in the paired samples–i.e., the samples having the same source language (Figure 3a-f, and Figure 3g-l)–even if involving different tasks. We observe a reordering phenomenon from the English audio "*cheap money*" and its Italian textual counterpart "*denaro a buon mercato*",[15] which is reflected in the saliency maps (Figure 3d and e) and also captured by cross-attention (Figure 3f). We also observe that there are some patterns captured by the attention that are not reflected in the input. For instance, in Figure 3f, the first words ("*È solo*") attend–albeit with relatively low scores–to the audio frames between 75 and 85, while this pattern this pattern is absent in the relevance scores of both the encoder output and the input. Consistent with the findings discussed throughout the paper, this example illustrates that while attention generally follows the saliency patterns identified by feature attribution, some discrepancies persist.

# F LIMITATIONS

This work provides an in-depth analysis of cross-attention explainability in encoder-decoder S2T models. While it yields actionable insights, some limitations should be acknowledged. First, our experimental scope is restricted to ASR and ST. Although these tasks are central to S2T-based AI (Radford et al., 2023; Barrault et al., 2023), we do not evaluate other downstream tasks such as spoken question answering or speech summarization, which may involve different dynamics in decoder attention. Second, our multilingual analysis is limited to English (a Germanic language) and Italian (a Romance language), due to the high computational cost of large-scale model training across a broader

---

[15]Same colors reflect the same concepts.

set of languages. Third, we focus on models trained from scratch but do not include architectures based on Speech Foundation Models (SFMs) paired with large language models (LLMs), often referred to as SpeechLLM–a recent growing area of interest in S2T modeling (Gaido et al., 2024b). As our analysis focuses on evaluation, our goal was to completely avoid data contamination issues (Sainz et al., 2023), which is a problem affecting almost every SFM and SpeechLLM architectures currently available, as we have no control over their training data, and, for this reason, we decided to retrain the models from scratch. Fourth, our analysis relies on SPES to compute reference explanations, acknowledging that, as an empirical method, it may introduce some margin of error. However, in the absence of a gold or human reference–which is unattainable in practice–we adopt SPES as a *silver* reference, since it represents the state of the art in explainability for speech-to-text. We further validate this choice in Appendix C, showing that SPES achieves very high quality explanations (deletion scores >90 on ASR and <3 on ST), making it a more faithful option than less robust alternatives from the generic XAI field such as gradient norms (Covert et al., 2021a)).

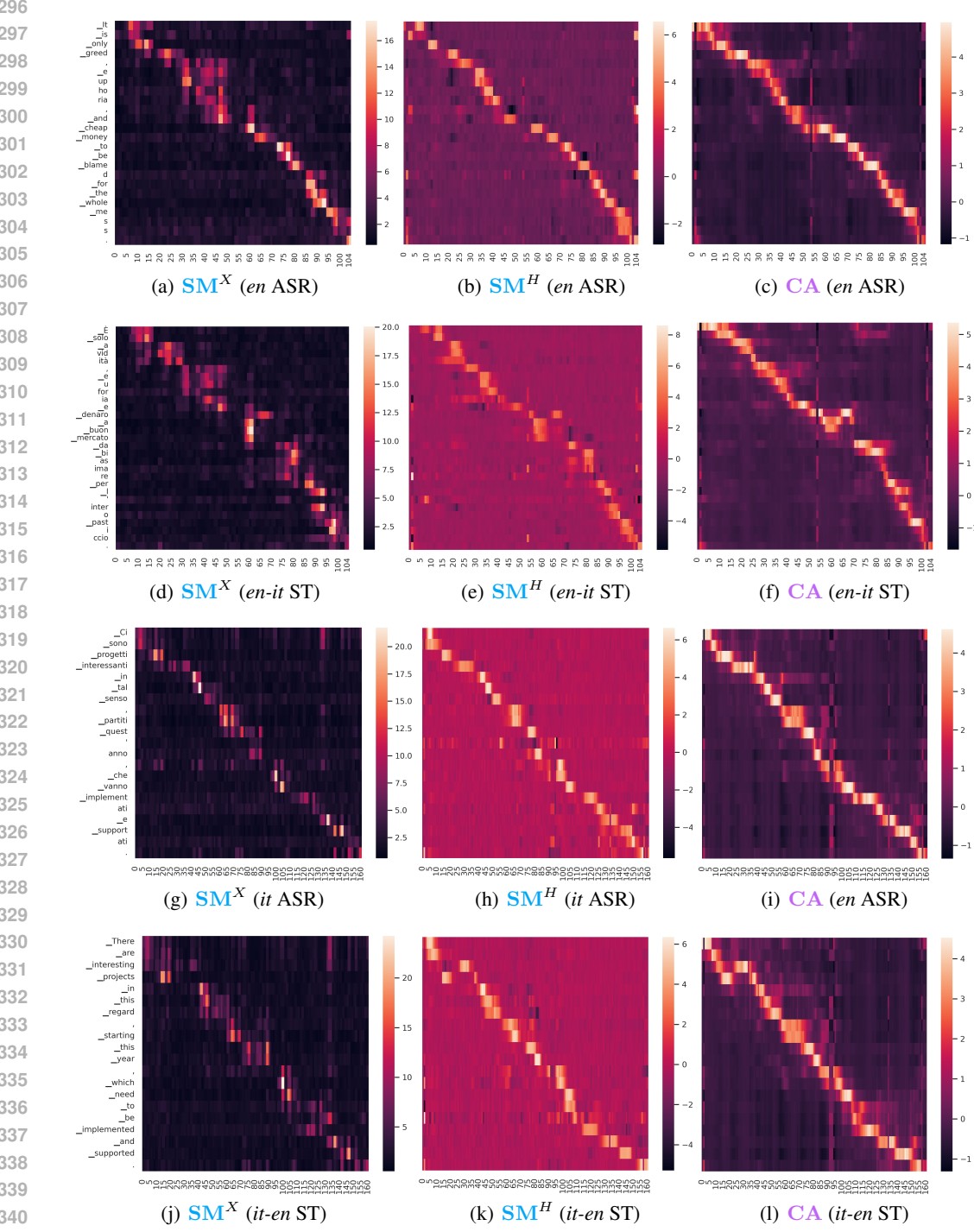

Figure 3: Example of input (first column) and encoder output (second column) saliency maps and cross-attention matrix (third column) produced by the `large` model for a paired *en* ASR (first row) and *en-it* ST (second row) sample, and a paired *it* ASR (third row) and *it-en* ST (fourth row) sample. The *en/en-it* reference sentence is "*Are only greed, euphoria and cheap money to be blamed for the whole mess?*" (English) and "*Avidità, euforia e denaro a buon mercato sono veramente le uniche cause di questo disastro?*" (Italian). The *it/it-en* reference sentence is "*Ci sono progetti interessanti in tal senso, partiti quest'anno, che vanno implementati e supportati.*" (Italian) and "*There are promising projects of this kind, to begin this year, which must be implemented and supported.*" (English).

