# OpenReview forum: "Cross-Attention is Half Explanation in Speech-to-Text Models"
_ICLR.cc/2026/Conference — ICLR 2026 Conference Withdrawn Submission_

### Official Review · Reviewer_gh84 · 2025-10-27

**Soundness:** 3
**Presentation:** 3
**Contribution:** 3
**Rating:** 6
**Confidence:** 3

**Summary:**

The paper addresses the question of whether or not cross attention values in encoder-decoder speech to text models (S2T) as explanations of the model's predictions. Since cross-attention weights are used in this domain in down stream tasks like time-stamp estimation, they argue that the claim that the attention weights reflect the alignment between the input and output must be validated. They use SPES to generate input relevance maps and use this as the ground-truth and compare it against the cross attention values. The analysis spans monolingual and multilingual models, ASR and ST tasks, and various model sizes. Their primary findings is that while the aggregate cross-attention score shows moderate to strong correlation, it only captures about 50% of the input relevance.

**Strengths:**

+ While in NLP there has been quite a few studies on whether or not attention values can be used as explanations, there has not been mucvh work on cross attention in S2T models. This is an important gap to fill, especially since these values are used to estimate other alignment and timestamping in practice.
+ comparing cross attention to both the input and the output saliency is useful, allowing one to assess both explanatory power of the CA, but also understand the effect of context mixing that happens in the encoder
+ training their own model from scratch prevents data contamination and ensures a cleaner evaluation.

**Weaknesses:**

- this study and its quantitative obervation (half) hinges on the use of the SPES as a proxy for ground  truth. Although this method is the current SOTA, the quantitative conclusions may change if something more powerful comes along. So, I am not sure that the title of the paper should be that specific.
- The choice of Pearson correlation is justified by saying that the magnitude matters; however, Pearson correlation assumes a linear relationship and can be sensitive to outliers. It would be better to see supplementary results on other types of correlations like Spearman, Kendall.
- one of the motivations of the work was that the cross-correlation is that it is used for downstream tasks like timestamp prediction. However, the effect of this partial correlation on errors in prediction have not been tested.

**Questions:**

1. Can you provide some ancillary results on other types of correlations as mentioned earlier?
2. You note that cross attention correlates better with output saliency than input saliency. However, even with the output saliency, the correlation is not perfect. Why might that be?
3. In the discussion you note that the cross attention achieves a deletion score of 41.2 compared to 52.9 for aggregated SM^X. This seems like it would be a more direct measurement of explanation quality than correlation. Was there a reason why these results were reported in the appendix instead of the main paper?

---

### Official Review · Reviewer_4cEB · 2025-10-30

**Soundness:** 3
**Presentation:** 3
**Contribution:** 2
**Rating:** 4
**Confidence:** 5

**Summary:**

The authors analyze whether decoder cross-attention in S2T models can serve as an explanatory proxy. They compare cross-attention matrices (aggregated across heads/layers) with saliency maps computed by SPES both at the input and encoder outputs levels. Correlations are reported across ASR/ST, English/Italian, and three model scales. Their findings indicate: (1) moderate–strong correlations only after aggregation (2) cross-attention explains about half of input relevance (best pearson correlation of 0.58 for input), and 52–75% of encoder-output saliency (3) averaging across heads/layers helps, with later decoder layers align better (4) deletion tests show attention heatmaps are weaker explanations than SPES. The authors conclude cross-attention is informative but incomplete as an explanation.

**Strengths:**

- Clear negative/positive result framing: attention is informative but incomplete.
- Breadth of analyses and empirical validation: ASR/ST, mono/multilingual, multi-scale with consistent trend that later decoder layers and aggregation across heads/layers increase alignment.
- Deletion tests support correlation findings: CA explanations underperform SPES, reinforcing that CA should be auxiliary not primary for XAI.

**Weaknesses:**

- While the paper offers an empirical study on the validity of cross-attention as an explanatory tool, and concludes that it should not be treated as a standalone XAI method, this conclusion is not surprising. It is already well established in the literature that (cross-)attention scores alone are insufficient for meaningful explanation. To derive real explanatory value, attention must typically be combined with gradient-based information or perturbation techniques. As such, the study largely confirms what is already known, and the conclusion lacks novelty or actionable insight. Therefore, both the empirical investigation and its relatively weak conclusion are unlikely to be of significant interest to the ICLR community. In my view this is the main weakness, but I am open to hear other thoughts.

- the global Pearson ρ on flattened maps lacks locality and uncertainty. S2T alignment is token-local and time-local. A global ρ can hide systematic misalignments (e.g., at word boundaries or under re-ordering in ST). Without intervals and per-token analyses, it’s hard to assess reliability for downstream tasks like timestamps and alignment.

- Relyingon SPES as the sole reference.while SPES is reasonable, relying on a single attribution family (perturbation-based) risks method-specific biases. Cross validating with one gradient-based method or a tiny human audit on a subset would increase confidence that 'half is not an artifact of SPES.

- Although the paper argues CA is cheap, the reference against which CA is judged is so expensive (generating SPES maps is expensive: 27 hours for base to 3-8 days for small/large on a single A40) that regular auditing becomes impractical. Recommendations about when CA is “good enough” vs. when SPES is required would help practitioners. E.g., given cost, when is CA-only adequate? Can you propose simple heuristics (e.g., last-layer + head averaging) and risk flags where SPES is recommended?

**Questions:**

Please see the weaknesses section.

---

### Official Review · Reviewer_Dycs · 2025-10-31

**Soundness:** 1
**Presentation:** 3
**Contribution:** 1
**Rating:** 0
**Confidence:** 5

**Summary:**

The paper’s main argument is that cross-attention scores only partially explain how an encoder–decoder model works on speech and cannot be used alone for model interpretation. The authors conduct a correlation analysis between the attention map and feature-attribution methods and find that the correlation is weak.

**Strengths:**

No

**Weaknesses:**

* From the abstract itself, the contribution is already minor. The debate over the explainability of attention scores is very old, and I don't think people usually believe that cross-attention scores can explain model behavior. As a result, the conclusion is expected, and the results presented are also expected. Either *“attention scores are in fact strongly explainable”* or *“attention scores are misleading and lead to serious issues such as …”* would constitute a meaningful contribution. However, the current conclusion is common sense; thus, the contribution is minor.

* People use cross-attention scores for timestamp segmentation *because it works*, not for interpretability. It would be a great contribution if the authors conducted further analysis on the segmentation and found that attention-based methods lead to systematic errors, but the authors only mention that “it might be wrong.”

* The methodology used in the paper is also problematic. They treat a very recent attribution method, *SPES*, as the “ground truth” for model interpretability. Why? It is non-standard and unconvincing. There are many well-known and standard attribution methods available; none of them alone are sufficient to represent model behavior. However, the paper only presents results comparing to a single ground truth, which makes the conclusion unreliable. I would expect to see at least two (three would be better) attribution methods and a study of the attention’s behavior by comparing them to determine whether there is a consistent conclusion across various attribution methods.

* The paper only presents a trivial conclusion without further demonstration of actionable next steps. They mention that, based on their analysis, a better time-segmentation algorithm can be developed. You should present the results to justify this claim—it is straightforward and would strengthen your contribution. Similarly, they propose regularizing the cross-attention score with the feature-attribution method to improve ASR. Why not actually present the ASR results? Given that the current contribution is too minor, adding further results derived from your conclusion would be helpful.

**Questions:**

No

---

### Official Review · Reviewer_cKCU · 2025-10-31

**Soundness:** 3
**Presentation:** 3
**Contribution:** 2
**Rating:** 4
**Confidence:** 4

**Summary:**

This paper offers an analysis on the interpretability power of the cross-attention mechanism in encoder-decoder Speech-to-text (S2T) models. The authors examine whether cross-attention weights can reliably indicate which parts of the input speech signal influence the model's predictions. They compare the alignment patterns extracted via cross-attention weights with saliency maps obtained via SPES, a perturbation-based state-of-the-art feature attribution method for S2T models. The paper evaluates the correlation between the two methods across various architectures (monolingual, multilingual, and multitask models) and scales, finding that cross-attention correlates moderately but not strongly with saliency-based explanations.

**Strengths:**

- The paper is clearly written and well structured. The methodology and experimental setup are easy to follow.
- The study includes an extensive evaluation across different model sizes, languages, and task types (ASR and ST).

**Weaknesses:**

- The study focuses exclusively on traditional encoder–decoder architectures, a design that has seen declining prominence with the rise of decoder-only architectures. As a result, the findings may have limited applicability to current model paradigms, reducing the paper's long-term impact.

-  The analysis implicitly treats SPES as a reliable ground-truth explanation. However, perturbation methods can themselves be noisy and capture only one dimension of model interpretability. Relying on a single attribution baseline risks overestimating the discrepancy (or agreement) between cross-attention and "true" saliency.

- The approach of averaging cross-attention scores across subsets of layers and heads is overly simplistic. Different heads are known to specialize in distinct roles. By averaging across heads the analysis can conflate functionally diverse attention patterns and obscure meaningful specialization effects.

- The study relies exclusively on raw attention weights as the explanatory signal. This overlooks alternative formulations, such as norm-based ([Kobayashi et al., 2020](https://aclanthology.org/2020.emnlp-main.574.pdf)), that have been shown to better capture the effective contribution of value vectors to the cross-attention heads' outputs. Since attention weights merely scale values, a low weight may still yield a strong influence if the associated value vector has large magnitude. Ignoring this interaction can lead to misleading conclusions about which input regions actually drive model predictions.

**Questions:**

Since SPES is computationally expensive (27 hours to several days), could cross-attention still serve as a pragmatic approximation despite its limitations?

---

### Note · Authors · 2025-11-18

I have read and agree with the venue's withdrawal policy on behalf of myself and my co-authors.